# A Rare Case of Familial Schwannomatosis Showing Intrafamilial Variability with Identification of a Shared Novel Germline *SMARCB1* Mutation

**DOI:** 10.3390/medicina58111592

**Published:** 2022-11-03

**Authors:** Jun Hyun Lee, Jae Seok Jeong, Kum Ju Chae, Yeon-Hee Han, So Ri Kim, Yong Chul Lee

**Affiliations:** 1Department of Internal Medicine, Jeonbuk National University School of Medicine, Jeonju 54907, Korea; 2Department of Internal Medicine, Research Center for Pulmonary Disorders, Jeonbuk National University School of Medicine, Jeonju 54907, Korea; 3Research Institute of Clinical Medicine of Jeonbuk National University-Biomedical Research Institute of Jeonbuk National University Hospital, Jeonju 54907, Korea; 4Department of Radiology, Jeonbuk National University School of Medicine, Jeonju 54907, Korea; 5Department of Nuclear Medicine, Jeonbuk National University Medical School, Jeonju 54907, Korea

**Keywords:** *SMARCB1*, familial schwannomatosis, segmental schwannomatosis, novel variant, intrafamilial variability

## Abstract

Schwannomatosis is characterized by the presence of multiple schwannomas without landmarks of NF2. It is considered the rarest form of neurofibromatosis (NF). Here, we report the first case of familial schwannomatosis with regard to the segmental/generalized phenotype, in which the proband and the daughter present a distinct phenotype in this classification. The proband presents a generalized, painless, extradural type of schwannomatosis, while the daughter shows a segmental, painful, intradural type of schwannomatosis. Whole-exome sequencing of the affected individuals revealed a shared novel *SMARCB1* gene mutation (c.92A > G, p.Glu31Gly) despite the clinical variability. We thus suggest two points in the diagnosis of familial schwannomatosis: The identified novel germline *SMARCB1* variant can be reflective of a phenotypical progression from a segmental to a generalized type of schwannomatosis, or an intrafamilial variability in inherited schwannomatosis, which was not reported in previous literature. The specific combination of somatic *NF2* mutations may be a major factor in regulating the severity and scope of the resulting phenotype in schwannomatosis.

## 1. Introduction

Schwannoma is a well-encapsulated, benign tumor that originates from Schwann cells and forms the sheath of somatic or autonomic nerve fibers [1]. Schwannomatosis is currently considered the rarest form of neurofibromatosis (NF), a genetic disorder that causes multiple tumors in nerve tissues, including the brain, spinal cord, and peripheral nerves [2]. Jacoby et al. suggested the first diagnostic criterion for schwannomatosis; the presence of multiple schwannomas without NF2 landmarks [3]. Currently, schwannomatosis is definitively diagnosed when the patient is: (1) > 30 years of age with two or more schwannomas (not intradermal) and has at least one schwannoma with histological confirmation with no evidence of bilateral vestibular schwannomas on brain MRI scan and no *NF* mutation; (2) a pathologically confirmed schwannoma or intracranial meningioma and one affected first-degree relative [4]. The incidence of schwannomatosis is 1 in 40,000–1.7 million people, which is the rarest among NF. 

Schwannomatosis is most frequently sporadic; however, approximately 20% of cases are estimated to be inherited (familial) [5]. Schwannomatosis can also present with multiple schwannomas located on one limb or on five or fewer contiguous segments of the spine, which was coined as segmental schwannomatosis by MacCollin et al. The pathophysiology of schwannomatosis is currently unknown, and the characteristics of segmental schwannomatosis are not well known [6].

Germline mutations in *SMARCB1* and *LTZR1* are known to cause schwannomatosis, accounting for up to 85% of familial cases of schwannomatosis. However, germline mutations in *SMARCB1* and *LTZR1* alone do not trigger schwannomas, and additional somatic mutations in *NF2* are required. Most schwannomatosis cases with germline *SMARCB1* mutations follow a four-hit/three-step tumorigenesis model. The first step required a heterozygous germline *SMARCB1* mutation, and the second step included the loss of heterozygosity (LOH) of 22q while removing the wild-type *SMARCB1* allele and one of the two *NF2* alleles. The final step requires somatic mutation of the other *NF2* allele on the chromosome harboring the germline *SMARCB1* mutation [7]. Here, we present a rare case of a family with schwannomatosis harboring distinct clinical manifestations: the proband with a generalized, painless, extradural schwannomatosis and a daughter with a segmental, painful, intradural type of schwannomatosis. A novel shared germline *SMARCB1* variant was identified.

## 2. Case Presentation

### 2.1. Clinical Presentation (Proband)

A 66-year-old woman presented with an incidental finding of a lung mass (not otherwise specified), accompanied by palpable masses in the hands, legs, and back. She had a history of surgery for schwannoma in 1998. Further history-taking revealed that this was a spinal schwannoma. She had experienced severe back pain prior to surgery. In 2007, painless neurogenic tumors were incidentally found in the upper and lower portion of the psoas muscle: an ovoid-shaped epidural mass measuring 3.7 cm, in the lower neck, and at the level of the third and fourth cervical vertebrae. No further surgery was performed. In 2013, she underwent excision surgery for incidental neurogenic tumors of the right shoulder and palm. The diagnosis of cellular schwannoma was confirmed via excisional biopsy. These were accompanied by multiple masses with cystic degeneration at the level of the seventh cervical vertebra and first thoracic vertebra. Both show a connection to the neural foramen. In 2014, she underwent an additional surgery for a palpable infraclavicular mass, which was later confirmed as an ancient schwannoma. Additional masses were found in the brachial plexus and right median nerve (Figure 1). With regard to family history, the proband had four children. Only her second daughter was diagnosed with a schwannoma. Other children have been reported to have no schwannoma. The details of the daughters are stated in Section 2.2. 

The clinical laboratory evaluations were normal. Physical examination revealed no hearing deficits. Posteroanterior chest radiography revealed a neurogenic tumor in the posterior mediastinum of the left first intercostal region. The patient also had a pleural or neurogenic mass in the left sixth intercostal region. The proband then underwent a bone scan using 99m-Tc-HDP, which showed increased uptake in the posterior portion of the left first rib, accompanied by a sternal fracture. She also had degenerative changes in the joints between the third and fourth lumbar vertebrae. On chest computed tomography scans, the mass initially discovered on chest radiography was shown to be connected to the neural foramen. Lymph node metastasis was excluded. Additional neurogenic tumors were found in the subcutaneous layer of the lumbar spine (L2), the right erector spinae muscle (L3–L4), the left erector spinae muscle (L5), and the left obturator internus muscle. The tumors were confirmed using magnetic resonance imaging (MRI). Brain magnetic resonance imaging (MRI) revealed no evidence of vestibular schwannoma. The proband was diagnosed with schwannomatosis. 

Endobronchial ultrasound-guided transbronchial needle aspiration (EBUS-TBNA) of the mass in the posterior mediastinum, ultrasound-guided biopsy of the back mass, and fine-needle aspiration (FNA) of the mass in the left supraclavicular lymph node were performed. Initially, in consultation with the Department of Laboratory Medicine, she was recommended to undergo an excisional biopsy of the lesions. However, decisions were withheld after consultation with the department of chest/cardiovascular surgery and neurosurgery. Video-assisted thoracoscopic surgical biopsy (VATS) was withheld; she underwent an excisional biopsy of the protruding subcutaneous mass on the back at the Department of Neurosurgery. An ancient schwannoma was pathologically confirmed.

### 2.2. Clinical Presentation (Daughter)

The patient (proband)’s daughter, a 45-year-old woman, came to seek medical attention because of her family history. She had received treatment from other hospitals prior to the visit. After a review of accessible medical records, we found that the patient had an operational history of spinal intradural extramedullary schwannoma on the thoracic spine (T10–T11). She had severe preoperative left flank pain. In 2015, she was re-admitted to the hospital because of severe left flank pain. Spinal MRI revealed a neurogenic tumor at the T11–T12 level. Other imaging modalities did not reveal any extraspinal schwannomas. Excisional removal of the spinal tumor was performed. Intradural extramedullary schwannoma with calcification was pathologically confirmed. In 2021, spinal MRI revealed an enhanced dot in the cauda equina (L1), suggestive of a migratory process (Figure 2). The patient did not report any pain. Physical examination revealed no hearing deficits. The daughter was clinically diagnosed with schwannomatosis. Phenotypic differences between the proband and the daughter are stated in Table 1.

### 2.3. Sequencing Analysis

Lymphocyte DNA was extracted from the whole blood of the proband and her daughter. From the samples, 5870 genes with 80,692 target exons and 839 SNV and indels were captured using the Celemics G-Mendeliome Diagnostic Exome Sequencing Panel (DES). We performed whole-exome sequencing of 100 bp paired-end sequencing on MGI DNBSEQ-G400 (GC Green Cross Genome, South Korea). Disease-related variants were identified via alignment with the human genome (GRCh37/hg19). The mean depth of coverage was 162.47X with a quality threshold of 98.9%. Informed consent was obtained from the proband and her daughter. The panel identified a novel heterozygous variant in SMARCB1 (NM_003073.5: c.92A > G, p.Glu31Gly) in the proband and her daughter (Figure 3). This led to the molecular (final) diagnosis of both the proband and her daughter with schwannomatosis. An in silico analysis was performed to evaluate the pathogenicity of this variant.

## 3. Discussion

This report presents the first case of familial schwannomatosis with regard to the segmental/generalized phenotype, in which the proband and the daughter present a distinct phenotype in this classification. They shared a novel germline *SMARCB1* mutation. The proband and the daughter showed distinct clinical characteristics from their initial observation during the hospital visit. The proband had a history of multifocal (non-segmental) painless extradural neurogenic tumors in various portions of the body, while the daughter had a history of segmental schwannomatosis with recurrent intradural extramedullary schwannomas only in the thoracic spine (T9–T12) with extreme pain prior to treatment, from 2008 until now, since her first operation. However, radiological findings from 2021 have revealed an enhancing dot in the lumbar region (L1), which could be suggestive of a migratory process. Further history taking revealed that the proband initially suffered from a painful, focal spinal schwannoma and that she underwent surgery in 1998, 9 years prior to the next incidental finding of painless schwannomas in the right psoas muscle and lower neck in 2007 (epidural, T3–T4). The proband had not experienced any pain after her operation in 1998. Regarding pain, the proband did not experience any pain, while the daughter expressed pain, which is suggestive of another point of intrafamilial variability. The clinical features of the affected individuals are presented in Table 1. We initially concluded that there are phenotypic differences between the affected individuals, but we could not rule out the scant possibility of a shared phenotypic progression from initial painful intradural segmental schwannomas to painless extradural, multifocal (non-segmental) schwannomas. The possibility of evolution from segmental to generalized schwannomatosis has been addressed in previous studies [6]. It is uncertain whether the *SMARCB1* variant identified in our study shows a variably expressed phenotypic spectrum, or a single clinical course that shows a phenotypic progression from segmental to generalized schwannomatosis. Since the daughter is relatively young to display a full manifestation of the inherited features, an extensive follow-up of our case may help verify this genotype-phenotype relationship and may serve to refine and expand the knowledge of the pathogenesis of schwannomatosis in general. To our knowledge, no familial schwannomatosis cases with regard to the manifestations of non-segmental, segmental schwannomas have been reported. Whole-exome sequencing was performed to search for germline mutations in the affected individuals, especially focusing on *SMARCB1* and *LTZR1*. 

The variant c.92A > G (p.Glu31Gly) in exon 1 of *SMARCB1* identified in our case has not been previously reported in public databases (gnomAD, LOVD, NHLBI GO Exome Sequencing Project, and KRGDB). However, another variant, c.92A > T (p.Glu31Val), was reported by Bacci et al. [8] in a family with multiple meningiomas associated with schwannomatosis. The proband in this case had no history of meningioma. In silico analysis revealed that this variant was probably damaging. (SIFT score = 0 (deleterious), PolyPhen score = 0.991 (probably damaging), and MutationTaster = D (disease-causing/probably deleterious)). According to ACMG criteria, since this variant was absent from controls in the population data (PM2), and multiple lines of computational evidence support a deleterious effect on the gene (PP3), this variant was classified as a variant of uncertain significance (VUS). However, in light of the patient’s phenotype and in silico results, we considered the newly identified variant to be clinically significant and potentially pathogenic. In terms of this change located in the penultimate base of *SMARCB1* exon, the possibility of this variant affecting the splice donor consensus sequence was considered. The splice prediction score of this variant c.92A > G was evaluated in these software (dbscSNV: high possibility-ada_score: 0.9999711, rf_score: 0.996; SpliceAI: low possibility-donor gain: 0.13; MaxEntScan: moderate possibility-alt: −0.182, ref: 6.172, diff: 6.354). Thus, although the splice prediction for this variant is conflicting, we could not rule out the possibility of its effect on splicing. 

Regarding pain, a previous study on pain and germline mutations in schwannomatosis, reported by Jordan et al., proposed that self-reported pain was significantly greater in patients with germline *LTZR1* mutations than in those with *SMARCB1* mutations [9]. They also stated that, while tumor location was not the primary driver of pain, total tumor volume was associated with self-reported pain. The affected individuals in our case had differing levels of self-reported pain; the proband did not experience any pain since her first operation, whereas the daughter experienced recurrent right flank pain from intradural spinal schwannomas. With the proband having a higher total tumor volume and the difference in penetration depth between the proband and the daughter (extradural vs. intradural), this case does not support the pain characteristics proposed by Jordan et al.. 

The major limitation of our report is that the somatic mutation of *NF2* in tumors of the proband and the daughter was not evaluated. Tumor samples were insufficient for analysis. Additional evaluation of somatic *NF2* mutations may provide some explanation for the differences between affected individuals. Further functional validation through molecular studies is needed to support our hypothesis.

Our case has an implication for the diagnosis of schwannomatosis in these terms: The identified germline *SMARCB1* variant can reflect phenotypical progression from a segmental to a generalized type of schwannomatosis or intrafamilial variability in inherited schwannomatosis, which has not been reported in the literature. The specific combination of somatic *NF2* mutations may be a significant factor in the regulation of disease severity (e.g., pain) and the scope of the resulting schwannomatosis phenotype. 

## 4. Conclusions

Although definitive conclusions cannot be drawn from a single case, this report presents the first case of familial schwannomatosis with regard to the segmental/generalized phenotype, in which the proband and the daughter present a distinct phenotype in this classification. A novel germline *SMARCB1* mutation was identified, which we suggest is a clinically significant factor in producing the differences between the affected individuals.

## Figures and Tables

**Figure 1 medicina-58-01592-f001:**
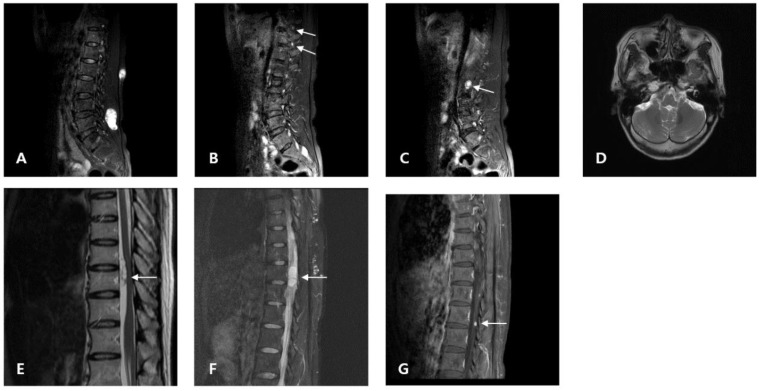
Proband (**A**–**D**): (**A**) T2-weighted sagittal MRI from 2021 showing a tumor (41.82 mm) with cystic degeneration in the lumbar level. Excisional operation was performed. (**B**,**C**) T2-weighted sagittal MRI from 2021 showing enhanced dots in the paraspinal connective tissue at the lumbar level (arrow). (**D**) Brain MRI showed no evidence of vestibular schwannomas. Daughter (**E**–**G**): (**E**) T2-weighted MRI from 2008 showing an intradural extramedullary tumor (T10–T11). (**F**) T2-weighted MRI from 2015 showing an intradural extramedullary tumor (T11–T12). (**G**) T2-weighted MRI from 2021 revealing an enhanced dot in the cauda equina (L1).

**Figure 2 medicina-58-01592-f002:**
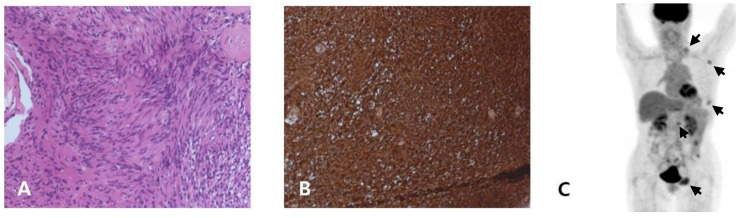
Proband (**A**–**C**): (**A**) Excisional biopsy (lumbar vertebra; Figure 2A) revealed tumor composed of spindle cells and Verocay bodies between palisading nuclei. (**B**) Immunohistochemistry revealed positive staining for S-100. (**C**) FDG PET/CT revealed mild FDG-avid masses.

**Figure 3 medicina-58-01592-f003:**
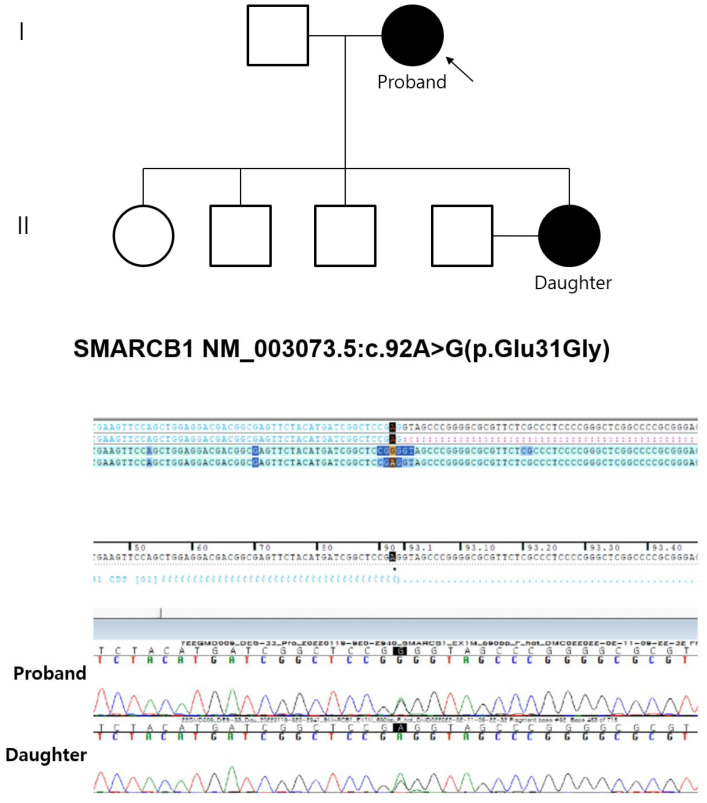
Pedigree and electropherogram of the proband and the daughter. The genotypes are indicated for the cases with available DNA. Black-filled: Affected individuals. Arrow: Proband.

**Table 1 medicina-58-01592-t001:** Phenotypic differences between the proband and the daughter.

	Proband	Daughter
Genotype/Phenotype	*SMARCB1* (c.92A>G, p.Glu31Gly)	
Ancestry	Korean	
Age at diagnosis	43	31
Segmental schwannomatosis	-	+
Vestibular nerve schwannomas	-	-
Spinal ependymoma	-	-
Spinal schwannoma	+	+
Extraspinal masses	+	-
Ocular lesions	-	-
Patient-reported pain	-	++
Intradural schwannoma	-	+
Meningioma	-	-

## Data Availability

All data regarding the findings are available within the manuscript.

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
