# Peer review of "A Rare Case of Familial Schwannomatosis Showing Intrafamilial Variability with Identification of a Shared Novel Germline SMARCB1 Mutation"

_medicina, 2022, doi:10.3390/medicina58111592_

Round 1

Reviewer 1 Report

Lee and collaborators describe an interesting case report of a familial schwannomatosis family with a novel, likely pathogenic variant in SMARCB1 gene. The case report is well written, with phenotypic description and extensive clinical evaluation of the two affected individuals. The results and the discussion are overall well-presented and framed. The rarity of this genetic condition and the novelty of the identified variant justify the interest in this case report.

Minor corrections:

- Gene symbols need to be italicized throughout the manuscript.

- All the figures need to be cited in the appropriated part of the result text.

- Lines 197 and 199: c:92A>G  and c:92A>T– correct to c.92A>G and c.92A>T

- Lines 204-206: the ACMG criteria (rules) used to classify this variant either as VUS or likely pathogenic should be stated, as well as the final classification. In example, for this novel variant, the following criteria could likely be applied: PM2, PP3, PM5, PP2 and PP4.

- The authors should discuss the possibility that, similar to the c.92A>T described by Bacci et al, the identified c.92A>G variant affects splicing. This change is located in the penultimate base of SMARCB1 exon and could affect the splice donor consensus sequence. Splice prediction software (e.g. spliceAI, dbscSNV and MaxEntScan) should be provided for the variant.

Author Response

Please see the attac

Point-by-point response

Dear Dr. Edgaras Stankevicius, Editor-in-Chief, and Dr. Sulman Basit, Guest Editor, and Dr. Asmat Ullah, Guest Editor,

We thank the editors and reviewers for their thoughtful comments on our work, and for the effort and time they put into reviewing our manuscript.

Here, we report the first case of familial schwannomatosis with regards to the segmental/generalized phenotype; in which the proband and the daughter present a distinct phenotype in this classification. The proband presents a generalized, painless, extradural type of schwannomatosis, while the daughter shows a segmental, painful, intradural type of schwannomatosis. Whole-exome sequencing of the affected individuals revealed a shared novel SMARCB1 gene mutation (c.92A>G, p.Glu31Gly) despite the clinical variability.

Several minor issues were raised during the initial review of the manuscript. We have addressed all these issues in response to the reviewers’ comments.

Response to Reviewer 1 Comments

  1. General comments

Lee and collaborators describe an interesting case report of a familial schwannomatosis family with a novel, likely pathogenic variant in SMARCB1 gene. The case report is well written, with phenotypic description and extensive clinical evaluation of the two affected individuals. The results and the discussion are overall well-presented and framed. The rarity of this genetic condition and the novelty of the identified variant justify the interest in this case report.

Reply: We really appreciate the reviewer’s contribution to the reviewing process and for the valuable comments on this case report.

  1. Specific comments:

Comment 1: Gene symbols need to be italicized throughout the manuscript.

Reply 1: Thank you for addressing this error. As the reviewer pointed out, we have corrected all the gene symbols in the revised manuscript.

Comment 2: All the figures need to be cited in the appropriated part of the result text.

Reply 2: As the reviewer pointed out, we have cited all the figures in the result text.

Comment 3: Lines 197 and 199: c:92A>G and c:92A>T– correct to c.92A>G and c.92A>T

Reply 3: Thank you for addressing the error, we have therefore corrected this sentence.

(Original Sentence) The variant c:92A>G (p.Glu31Gly) in exon 1 of SMARCB1 identified in our case has not been previously reported in public databases (gnomAD, LOVD, NHLBI GO Exome Sequencing Project, and KRGDB). However, another variant, c:92A>T (p.Glu31Val)…

  • (Corrected Sentence) The variant 92A>G (p.Glu31Gly) in exon 1 of SMARCB1 identified in our case has not been previously reported in public databases (gnomAD, LOVD, NHLBI GO Exome Sequencing Project, and KRGDB). However, another variant, c.92A>T (p.Glu31Val)…

Comment 4: Lines 204-206: the ACMG criteria (rules) used to classify this variant either as VUS or likely pathogenic should be stated, as well as the final classification. In example, for this novel variant, the following criteria could likely be applied: PM2, PP3, PM5, PP2 and PP4.

Reply 4: Thanks for the thoughtful and constructive comment. As indicated, we carefully reviewed how we classified this variant. We thus stated the PM2, PP3 criteria that was applied for this variant, and the final classification. (a variant of uncertain significance (VUS))

(Original sentence): According to ACMG criteria, this variant may be interpreted as a variant of uncertain significance (VUS).

  • (Corrected sentence): According to ACMG criteria, since this variant was absent from controls in the population data (PM2), and multiple lines of computational evidence support a deleterious effect on the gene (PP3), this variant was classified as a variant of uncertain significance (VUS).

Comment 5: The authors should discuss the possibility that, similar to the c.92A>T described by Bacci et al, the identified c.92A>G variant affects splicing. This change is located in the penultimate base of SMARCB1 exon and could affect the splice donor consensus sequence. Splice prediction software (e.g. spliceAI, dbscSNV and MaxEntScan) should be provided for the variant.

Reply 5: We express our gratitude for pointing out the integral content that needed to be filled. As the reviewer pointed out, we stated the splice prediction scores.

(Original sentence): None

  • (Corrected sentence): In terms of this change located in the penultimate base of SMARCB1 exon, the possibility of this variant affecting the splice donor consensus sequence was considered. The splice prediction score of this variant c.92A>G was evaluated in these softwares. (dbscSNV: high possibility - ada_score: 0.9999711, rf_score: 0.996; SpliceAI: low possibility - donor gain: 0.13; MaxEntScan: moderate possibility – alt: -0.182, ref: 6.172, diff: 6.354) Thus, although the splice prediction for this variant is conflicting, we couldn’t rule out the possibility of its effect on splicing.

Once again, thank you for your time and consideration. We hope that the revised manuscript is now satisfactory for publication in Medicina and we look forward to hearing good news from you.

Thank you for kind consideration.

Very sincerely yours,

Junhyun Lee, B.A.

Department of Internal Medicine, Jeonbuk National University Medical School, San 2-20, Geumam-dong, Deokjin-gu, Jeonju, 54907, South Korea.

E-mail: leejunhyun8@gmail.com

Yong Chul Lee, M.D., Ph.D.

Department of Internal Medicine, Jeonbuk National University Medical School, San 2-20, Geumam-dong, Deokjin-gu, Jeonju, 54907, South Korea.

Phone: 82-63-250-1664; Fax: 82-63-254-1609; E-mail: leeyc@jbnu.ac.kr

hment.

Reviewer 2 Report

This is a very solid case study regarding the SMARTCB1 role  in familial schwannomatosis. The quality of data and presentation is solid. My only note is to use HUGO nomenclature so italics for gene names.

Author Response

Dear Dr. Edgaras Stankevicius, Editor-in-Chief, and Dr. Sulman Basit, Guest Editor, and Dr. Asmat Ullah, Guest Editor,

We thank the editors and reviewers for their thoughtful comments on our work, and for the effort and time they put into reviewing our manuscript.

Here, we report the first case of familial schwannomatosis with regards to the segmental/generalized phenotype; in which the proband and the daughter present a distinct phenotype in this classification. The proband presents a generalized, painless, extradural type of schwannomatosis, while the daughter shows a segmental, painful, intradural type of schwannomatosis. Whole-exome sequencing of the affected individuals revealed a shared novel SMARCB1 gene mutation (c.92A>G, p.Glu31Gly) despite the clinical variability.

Several minor issues were raised during the initial review of the manuscript. We have addressed all these issues in response to the reviewers’ comments.

Response to Reviewer 2 Comments

  • Comments

This is a very solid case study regarding the SMARCB1 role in familial schwannomatosis. The quality of data and presentation is solid. My only note is to use HUGO nomenclature so italics for gene names.

Reply: We really appreciate the reviewer’s contribution to the reviewing process and for the valuable comments on this case report. Thank you for addressing this error. As the reviewer pointed out, we have corrected all the gene names in the revised manuscript.

Once again, thank you for your time and consideration. We hope that the revised manuscript is now satisfactory for publication in Medicina and we look forward to hearing good news from you.

Thank you for kind consideration.

Very sincerely yours,

Junhyun Lee, B.A.

Department of Internal Medicine, Jeonbuk National University Medical School, San 2-20, Geumam-dong, Deokjin-gu, Jeonju, 54907, South Korea.

E-mail: leejunhyun8@gmail.com

Yong Chul Lee, M.D., Ph.D.

Department of Internal Medicine, Jeonbuk National University Medical School, San 2-20, Geumam-dong, Deokjin-gu, Jeonju, 54907, South Korea.

Phone: 82-63-250-1664; Fax: 82-63-254-1609; E-mail: leeyc@jbnu.ac.kr
